# Cultural Use and the Knowledge of Ethnomedicinal Plants in the Pülümür (Dersim-Tunceli) Region

**DOI:** 10.3390/plants13152104

**Published:** 2024-07-29

**Authors:** Ahmet Dogan

**Affiliations:** Department of Pharmaceutical Botany, Faculty of Pharmacy, Marmara University, 34722 Istanbul, Turkey; adogan@marmara.edu.tr

**Keywords:** folk medicinal plants, Pülümür, Dersim-Tunceli, Anatolia, Zazaki, Kurdish

## Abstract

Pülümür has been a refuge place for internal and external exiles several times throughout history, and today it is a district of the province of Tunceli (known as Dersim in the region), which differs significantly from many settlements around it in terms of biodiversity and social aspects. The aim of this study is to identify, catalog, and report the traditional medicinal uses of plants in this province, where every living thing and many natural objects around it are regarded as sacred, with a scientific ethnobotanical approach. The field study was completed between May 2016 and July 2023. The collection of plant excursion and plant usage information was implemented in two stages; in the first stage, a survey about the collection and the uses of medicinal plants was conducted. At this stage, plant samples were collected by visiting 49 villages of the district and performing face-to-face interviews with 112 participants (51 female and 61 male). In the second stage, the usage statistics related to the data obtained from the field studies were determined. For this reason, every informant was interviewed at least twice, people who were previously interviewed were included too. In the course of this study, interviews from 211 participants (95 female and 116 male) were analyzed. As a result of analysis, the traditional medicinal uses of 120 plants belonging to 48 families were identified. The most taxa are identified from the Asteraceae (15), Rosaceae (13), Fabaceae (8), Lamiaceae (8), and Apiaceae (6) families. In Pülümür, these medicinal plants are mostly used for the treatment of wounds, abdominal pain, constipation, and diabetes.

## 1. Introduction

Medicinal plants have been employed for centuries in the treatment of various ailments and serve as the foundation for numerous modern pharmaceuticals. The study of these plants can lead to the discovery of novel drugs and therapies, offering potential cures or treatments for diseases that exhibit resistance to current medications. Indigenous communities often possess valuable ethnobotanical knowledge, meticulously passed down through generations. Documenting this knowledge not only helps in preserving cultural heritage but also ensures that such wisdom is not irrevocably lost as older generations pass away. By comprehending the applications of local plants, communities can develop sustainable practices that enhance health and well-being without depleting natural resources. This ethnobotanical knowledge can also bolster local economies, particularly in rural areas, through sustainable harvesting and the development of herbal products [1,2,3,4,5,6].

Turkey has a remarkable biodiversity due to its geographical location, climate, water resources, and geomorphological diversity. The richness of biodiversity in plants is determined by the number of endemic plants as well as the number of plant taxa growing in that region. The Anatolian Diagonal is one of the most distinctive biogeographic features for understanding the biological diversity of Anatolia. According to the first volume of the 11-volume series, “*The Flora of Turkey and the East Aegean Islands*”, this diagonal is defined by the geographical distribution patterns of plant species. The Anatolian Diagonal stretches from the northeast of Turkey, near the cities of Bayburt and Gümüşhane, to the southwest, dividing into two branches at the Central Taurus Mountains and the Nur Mountains as it approaches the Mediterranean. Notably, approximately 15% of Turkey’s endemic plant species are exclusively distributed either to the west or east of this biogeographic boundary [7,8,9,10]. This study was conducted in the Munzur Mountains which is a part of this diagonal and is located in the Pülümür region [11,12,13]. Pülümür is situated in the northeast of the deep valleys of Dersim (Tunceli), a city in the eastern Anatolian region of Turkey. The region’s social and cultural life is shaped by migrations from various cultures and economic activity centered on its mountainous terrain.

Anatolia, which has various natural resources including biodiversity, has been inhabited for thousands of years and is strongly affected by human–nature relations [14]. The function of plants in human life, which started with nutrition and shelter in the archaeological record, continued with the development of modern humans, adding medicinal uses, and today plants and humans are connected to each other by strong ties that cannot be broken [15]. Archaeological findings about the use of plants in Anatolia and Central Asia show that people began to use them as a food source [16], and then they started to use plants for medicinal purposes. One of the earliest scientific records on the use of medicinal plants is “*De Materia Medica*” by Dioscorides. This work, which originated in Anatolia, made a significant contribution to the development of modern medicine in Europe and the rest of the world [17].

The use of plants, considering the current COVID-19 epidemic, may be a focus of attention again in the future, especially for the discovery of new drugs, as well as being a complementary and preventive therapy [18,19]. Today, the use of wild plants for medicinal purposes is placed under stronger protection, especially in rural and relatively isolated communities, due to natural, ethnic, political, and economic reasons. Accordingly, ethnobotanical research is more productive in communities that have preserved traditional knowledge about the use of plants for medicinal purposes [20,21,22,23].

Minorities living in Anatolia in various periods of history have verbally transferred their cultural traditions from generation to the next generation, despite sovereign governments’ imposition of their own values and contempt for, exile of, and even mass murders of local peoples. The region known today as Tunceli has been under the influence of many pressures and security policies throughout history. It has received great migrations of people and has been a settlement area where different cultures were settled by governments in order to be easily managed. The most devastating effect in recent history occurred in 1937–1938, when thousands of people died and as many were forced to migrate as a result of the disagreements between the central Turkish government and some of the Dersim tribes of the period, regarding the dominance of the region. In this process, the name of the Dersim region was changed to Tunceli by law [24]. Despite this, most of the people living in the region, where historical and cultural awareness is strong, still use the name Dersim, not Tunceli. Today, the province of Dersim still differs from the central government’s approach and even from all other local governments by having the only communist local government in Turkey.

Written official population records about the demographics of Tunceli starting from 1518 show that various ethnic groups such as Armenians, Turkmens, Kurds, Zazas, and religious minorities inhabited this region. According to these records, many ethnic and religious groups migrated to these lands at various periods of history or were forced to migrate from these lands including during the Ottoman–Iranian war. A very cosmopolitan cultural richness has been formed with these migrations, and traces of many traditions belonging to these cultures are still observed in the Pülümür region. The impact of these migrations and the mountainous terrain, which allowed only limited agriculture, has paved the way for the continued expansion of animal husbandry [24,25,26].

Livestock has offered a semi-nomadic life which is intertwined with nature. Such a lifestyle has nurtured the beliefs of sacredness of land–water and nature, creating a cult. This history highlights how migration, geography, and historical events have shaped the unique cultural and social landscape of Tunceli, influencing its traditions and economic practices. Anthropologically, this diverse and turbulent history underscores the resilience of the region’s inhabitants and their ability to maintain and adapt their cultural identity amidst external pressures and changing political landscapes. Dersim is recognized as a significant center of Anatolian Alevism, known for maintaining its unique cultural and religious characteristics over centuries. The population primarily speaks Kurdish, Zazaki, and Turkish, reflecting a diverse ethnic composition that transcends conventional ethnic and religious categorizations. Unlike the traditional definitions of Alevism, which often center around Turkish identity, the people of Dersim emphasize their distinctiveness by referring to their beliefs as “Raa Haq” (the path of right or justice), highlighting a unique ethno-religious identity [27]. This distinct identity results from the region’s historical and geographical context. Dersim’s rugged and isolated terrain has historically provided a refuge for diverse ethnic and religious groups, allowing the preservation of unique traditions and practices. The synthesis of various beliefs and ethnicities within Dersim has created an ethno-religious structure that is different from other regions in Turkey and neighboring countries. The historical continuity of Alevism in Dersim, shaped by both distant and recent historical events, underscores the region’s unique cultural and religious landscape [28,29].

This research study focused on compiling ethnomedicinal information of plants in the Pülümür region, transforming the oral knowledge of traditional usage of medicinal plants into a scientific written report, including the evaluation of traditional uses of plants according to age, gender, and education level and comparing traditional uses of plants with other similar regions to reveal new ethnomedicinal uses of plants. The following research questions guided this study:What are the specific traditional medicinal uses of plants in Pülümür?Which plant species are commonly employed, how are these plants traditionally prepared and administered, and what ailments or health conditions are treated with these medicinal plants?How do different demographic groups (e.g., age, gender, and education level) in this region vary in their knowledge and use of medicinal plants?Are there unique or novel ethnomedicinal uses of plants in Pülümür compared to those in similar neighboring regions?

## 2. Results

During the field studies in Pülümür, 652 plant samples were collected, and the information on 120 taxa used in traditional treatment is listed alphabetically according to family and genus names in Table 1.

Among the 120 plant taxa belonging to 48 families, only 2 of them are cultivated plants. The most commonly used medicinal plants are Asteraceae (15 taxa), Rosaceae (13 taxa), Fabaceae (8 taxa), Lamiaceae (8 taxa), Apiaceae (6 taxa), Fagaceae (5 taxa), Amaranthaceae (4 taxa), and Poaceae (4 taxa) family members (Figure 1).

Interviews were conducted with a total of 211 informants, 95 (45%) females and 116 (55%) males. The ages of informants ranged between 18 and 89 years. About 90% have graduated from at least one level of school, and 28% have graduated from college. It was observed that females were more competent in the use of herbs than males. The demographic information of the informants is presented in Table 2.

In the Pülümür region, there are significant variations in the level of knowledge about traditional folk remedies across different age groups. Specifically, women over the age of 45, particularly those with relatively low education levels, demonstrate a substantially higher awareness and understanding of the health-related uses of local plants. This demographic is notably more knowledgeable about the medicinal applications of several species. For instance, they are well-versed in the use of Pistacia species for treating mouth sores, *Berberis crataegina* for hemorrhoid treatment, Euphorbia species for wart treatment, Alcea species for alleviating coughs, and *Eremurus spectabilis* as an antipyretic. In contrast, younger generations, especially those under the age of 25, exhibit a marked deficiency in this traditional knowledge.

This study indicates that younger individuals, irrespective of their gender, generally know little about the aforementioned medicinal uses. This lack of knowledge is particularly pronounced among males under 25 with higher education levels, many of whom are unfamiliar with the local names of most plants used traditionally in the region. This suggests a significant generational gap in the transmission of ethnobotanical knowledge.

However, this study also notes an interesting trend among younger individuals with higher education levels. While they lack knowledge of traditional uses specific to the Pülümür region, they demonstrate a better understanding of the medicinal uses of plants that are widely recognized in neighboring regions. This demographic shows greater familiarity with the uses of species from the genera Hypericum, Anchusa, Urtica, Populus, and Rubus. These plants are more commonly utilized in the broader areas surrounding Pülümür, indicating that formal education may be exposing younger generations to a different set of ethnomedical knowledge that is more regional rather than local.

In our field studies, the decoction, infusion, eaten fresh, crushed in stone mortar, mixed with butter, chewing, maceration, peeling, burned on a hot iron, dried, etc., methods were mentioned among the traditional medicine methods. The most commonly used forms of preparation are decoction (34), infusion (31), fresh (20), crushed in hand-made stone mortar (Figure 2) (12), and mixing with butter (5). In addition, application methods of these uses are internal (113) and external (52). The external use of latex-bearing plants was particularly emphasized. Oleates are prepared by mixing these herbs with butter. It was stated that stone mortars and grinders were especially selected for grinding and pulverizing the plants, and, thanks to this, the effects of the plants were used in the treatment without loss.

It was observed that most of the aerial parts (71) are used in traditional medicine. However, parts such as the fruits (37), leaves (36), root (17), latex (15), and seed (12) are frequently used alone, too (Figure 3).

According to the UV data, the most commonly used plants, regardless of the age of the informant, are *Allium tuncelianum*, *Tragopogon buphthalmoides*, *Tragopogon reticulatus*, *Helianthus tuberosus*, *Rosa canina*, *Stachys lavandulifolia*, *Chaerophyllum crinitum*, *Prunus trichamygdalus*, and *Rheum ribes*. Apart from those mentioned above, species such as *Gundelia*, *Hordeum*, and *Juglans*, which are used as food or spice, are also used as medicinal plants extensively.

Asteracea members carrying latex are frequently used in wound treatment. The use of oleate made with butter of Scorzonera species in the treatment of skin diseases has been used for generations, including for war wounds. It was observed that many of the participants kept this oleate at home. Female participants over the age of 60 who contributed to our research in the region stated that they had frequently used the *Arum rupicola* herb in the past for contraception purposes.

Pülümür’s traditional medicinal plants are utilized to treat 53 various types of human diseases or disorders. Herbal treatments are most commonly used in the community to treat digestive system disorders (67), skin diseases (35), the respiratory system (22), nervous system disorders (14), endocrine system disorders (16), urogenital system disorders (10), and cardiovascular system disorders (10) (Figure 4).

## 3. Materials and Methods

### 3.1. Study Area

Pülümür district is located in the northeast corner of Dersim (Tunceli) province, in the Upper Euphrates Section of the Eastern Anatolia Region, and is adjacent to Bingöl to the east, Ovacık to the west, Erzincan to the north, and Nazimiye to the south (Figure 5). Its area is 1476 km^2^. Although it is equidistant from the provincial centers of Erzincan and Dersim (Tunceli), it is difficult to reach this district because of steep mountain roads which are off the main transportation roads. The Karagöl Mountains rise in the south of the district, the Bağırpaşa Mountain is in the northeast, and the Munzur Mountains (Figure 6) are in the northwest. Pülümür was founded in a narrow valley in the north–south direction opened by the Pülümür stream [26]. Pülümür consists of 49 villages spread over different altitudes starting from 1500 m above sea level and rising to 2300 m above sea level. On the summit of these mountains, there are Navgöl, the Buyer Baba Lake, the Kırdım village lakes, and the Bağır Paşa mountain crater lakes, as well as the highland areas where nomadic animal husbandry is still carried out by the Kurdish-speaking Şavaklı and Zaza-speaking Zazaki communities who take their goat and sheep herds during the spring and summer months and stay there during this period.

In the region where the continental climate is dominant, the annual average temperature between 1960 and 2021 was measured as 12.7 °C. The highest average was 43.5 °C in July, and the lowest temperature was measured in January at −30.3 °C. The annual average precipitation is 872.2 mm, and the average number of rainy days is 102.2 [70].

The vegetation in this region is quite diverse, with different types of plants found in various habitats such as forests, riverbanks, slopes, and forest clearings. The most common forest cover consists of pure or mixed oak assemblages (*Quercus brantii*, *Q. libani*, *Q. peraea*, *Q. robur*, *Q. infectoria*, and *Q. petraea subsp. pinnatiloba*) along with juniper species (*Juniperus communis subsp. nana*, *J. excelsa*, *J. foetidissima*, and *J. oxycedrus subsp. oxycedrus*) and sparse forests of aspen (*Populus tremula*). Along riverbanks and slopes, the most prevalent species are willow (*Salix alba*, and *S. capraea*), walnut (*Juglans regia*), redwood (*Alnus glutinosa*), and fig (*Ficus carica* subsp. *rupestris*). In forest clearings, species such as *Pistacia eurycarpa*, Celtis *tournefortii*, *Prunus divaricata* subsp. *divaricata*, *Prunus divaricata* subsp. *ursina*, *Pyrus syriaca* var. *syriaca*, *Crataegus meyeri*, *Crataegus monogyna* subsp. *monogyna*, *Crataegus pseudoheterophylla*, and *Berberis crataegina* are common. Additionally, plants in the form of trees or shrubs such as *Daphne oleoides* subsp. *oleoides*, *Rosa canina*, and *Rubus sanctus* are frequently encountered. This diversity highlights the complex and varied ecosystems within the mountainous areas in the north of the region, with each species adapted to specific environmental conditions [24].

Although there is no detailed study on the flora of Pülümür, floristic research has been carried out on the Munzur Mountains, which is the natural border of the county in the north. In the study, 1407 taxa were identified from the Munzur Mountains, and it was determined that 45.7% of the flora were of Iranian–Turanian origin, 8% European–Siberian, and 4.4% Mediterranean. It has been determined that 39 of the 275 endemic flora found in the Munzur Mountains are found only in the Munzur Mountains [71].

### 3.2. Interviews about Traditional Medicinal Plant Usage and Statistical Data Collection

This study was carried out in two stages and a total of 121 days between May 2016 and July 2023. The first phase of this study involves a botanical collecting expedition, during which various plant species are gathered and their potential uses identified. This stage is critical for compiling a comprehensive inventory of plants and documenting their medicinal traditions. In the next phase, quantitative data are collected through structured interviews with each participant in this study. These participants are systematically questioned about their knowledge and uses of the identified plants. This approach facilitates a robust analysis of plant use patterns in the studied population, providing statistically relevant information. In the region, characterized by a very low population density, primary informants were identified with the assistance of non-governmental organizations and local administrators (such as the Mayor and Mukhtar). The pool of volunteer participants was expanded based on the recommendations provided by these initial interviewees.

In the first stage, 49 villages and centers of the Pülümür district were visited for plant excursion and collecting information about plant usage, which lasted 89 days between 2016 and 2018. Plant samples were mostly collected during nature walks with people who knew the region and the uses of these plants. All the plants discussed in the results section were gathered during this phase. The collected plants were not only identified by the individuals who helped collect them but were also verified with other participants encountered on the days of the field studies. During this phase, every participant was interviewed at least once, and the questionnaire in Appendix A was administered to them. However, due to the varying schedules of field studies conducted on different days, it was not possible to show every plant to each participant at least once. This part of the research included a pilot study focusing on the use of plants, ensuring a comprehensive understanding of the ethnomedicinal knowledge present in the community.

In the second stage, face-to-face interviews were conducted over 32 days during 2019. In this stage, more efficient and accurate statistics related to the data obtained from the first stage are determined. The questionnaire in Appendix B was used in the second stage of this study. Due to the COVID-19 pandemic, 133 interviews were conducted online during 2020–2023. Since old people have difficulty using the technology, online interviews were mostly conducted with the help of children and/or grandchildren accompanying the elder participants. During the statistical data collection procedure, at least two interviews were conducted with each participant.

The purpose of this research was explained to all interviewees, the ethical rules were considered, and informed consent forms were signed by each participant [72,73]. Most of the interviews were conducted in Turkish, and the Zazaki and Kurdish interviews were mostly conducted with elderly people who could speak limited Turkish. Most of the plant names were recorded with Zazaki and Kurdish local names, even if the interview was in Turkish. No translator was used during the interviews and all interviews were conducted by the author.

The ethical implications of this study were subjected to rigorous and meticulous examination throughout the research process. All requisite permissions were obtained from the Office of Human Subjects prior to the commencement of data collection. All participants were fully informed about the nature of this study, and their consent was obtained in accordance with the ethical standards set out by the Office of Human Subjects. The scope of this project was elucidated for all participants, and their right to withdraw at any point in this study was underscored. Each participant was informed that their identities, personal data, and the information provided would be kept strictly confidential and would not be shared in any way. These measures were implemented in accordance with the ethical standards and institutional guidelines of Marmara University’s Human Subjects Office, thereby ensuring that the privacy and confidentiality of all individuals involved in the research process are protected.

### 3.3. Plant Materials

The plant samples were collected over 89 days between May 2016 and September 2019. Growing periods and the height of the villages above sea level were considered in the collection of plants. Plants were mostly collected in villages with people who knew the plants. Plants were described using *Flora of Turkey and the east Aegean Islands* [7,74,75] and *Illustrated flora of Turkey* [8]. While the information about the plants was being compiled, the plant specimens themselves or their pictures were shown to the interviewees. Plant samples are kept in Marmara University Faculty of Pharmacy (MARE) Herbarium. Scientific current names of plant taxa were written according to the Turkish flora checklist [9] and *The Plant List* website [76].

### 3.4. Statistical Data

Statistical data analysis in this study was conducted following guidelines for best practices in ethnopharmacological research [77]. From these guidelines, the most appropriate data presentation methods for our field of study were selected. Obtaining information about the use of plants was carried out in two stages, and each informant was asked about the use of each plant separately. For this reason, it was deemed appropriate to only calculate use value.

Use Value (*UV*)

The *UV* value was created to quantify the use of plants in this study [78,79]. The calculation of this value has been updated to be calculated with the following formula with less errors when there are enough 
informants [80].
$$UV_{s}=\frac{\sum_{i=1}^{n}U_{is}}{n_{s}}$$

In this formula, “*U_is_*” equals the number of medicinal uses of taxa mentioned by the informant “*i*” and the “*n_s_*” is the number of informants interviewed for species *s.*

## 4. Conclusions

Findings of this study are compared with other studies conducted with similar ethnic groups or in a proximate location. As a result, studies in the provinces of Turkey’s eastern region and neighboring countries, as well as cities in Iran and Iraq with similar cultures, were compared [30,31,32,33,34,35,36,37,38,39,40,41,42,43,44,45,46,47,48,49,50,51,52,53,54,55,56,57,58,59,60,61,62,63,64,65,66,67,68,69,81,82]. The following taxa are reported as medicinal plants extensively in all of these studies: *Urtica dioica*, *Rosa canina*, *Mentha longifolia*, *Malva neglecta*, *Rheum ribes*, *Juglans regia*, *Teucrium polium*, *Glycyrrhiza glabra*, and *Plantago major*.

In conclusion, results demonstrated that plants continue to play an important role in people’s basic health care in the Pülümür district. The use of 120 plant taxa belonging to 48 families for 53 types of disorders and diseases in humans were determined. In the Pülümür population, the most common herbal remedies are utilized for digestive system problems, skin illnesses, the respiratory system, neurological system disorders, endocrine system disorders, urogenital system disorders, and cardiovascular system disorders. *Scorzonera veratrifolia*, *Scorzonera tomentosa*, *Allium tuncelianum*, *Tragopogon buphthalmoides*, *Tragopogon reticulatus*, *Helianthus tuberosus*, *Rosa canina*, and *Stachys lavandulifolia* are the most widely utilized plants, accordingly their Use Values were found to be very high. This study showed that the most extensively used taxa in this district for medicinal purposes were *Urtica dioica*, *Rosa canina*, *Mentha longifolia* subsp. *typhoides*, *Malva neglecta*, *Rheum ribes*, *Juglans regia*, *Teucrium polium*, and *Glycyrrhiza glabra*, similar to the other Kurdish eastern districts of Turkey and some Iranian and Iraqi cities. The aerial parts of plants are mostly applied in traditional medicine. Along with that, the fruits, leaves, root, latex, and seed are all widely used by themselves. The taxa carrying latex are frequently used in wound treatments. The use of oleate made with butter and the Scorzonera, Tragopogon, and Pistacia species in the treatment of skin diseases has been used for generations, including for wounds and burns. Many of the participants always have this oleate at home ready to use in case of emergencies as well as for daily use.

As a result of comparative analysis with studies considering cultural and geographical closeness, local or rare uses of 23 different taxa were identified. These taxa are *Amaranthus albus*, *Allium cardiostemon*, *Allium rotundum*, *Chaerophyllum crinitum*, *Heracleum cyclocarpum*, *Heracleum platytaenium*, *Echinops viscosus* subsp. *bithynicus*, *Helichrysum rubicundum*, *Tanacetum cilicicum*, *Tragopogon buphthalmoides*, *Astracantha amblolepis*, *Cicer bijugum*, *Lathyrus roseus*, *Trifolium hybridum*, *Dactylorhiza osmanica*, *Papaver pseudo*-*orientale*, *Setaria viridis*, *Ranunculus fenzlii*, *Cotoneaster ellipticus*, *Prunus trichamygdalus*, *Sorbus umbellata*, and *Tamarix tetrandra*. Genera such as Amaranthus, Allium, Chaerophyllum, Heracleum, Echinops, Helichrysum, Tragopogon, Cicer, Dactylorhiza, Prunus, and Sorbus are commonly used as food or tea, which can influence the digestive system. It is noteworthy that *Tragopogon buphthalmoides* and other members of the latex-containing Asteraceae family are widely utilized for external wound treatment. Similarly, Euphorbia species, also latex-producing plants, are applied externally, indicating an awareness of their toxic effects and thus avoidance of internal use. The utilization of *Astracantha amblolepis* and *Echinops viscosus* subsp. *bithynicus* as aphrodisiacs and for male health issues is significant. These plants are also collected as animal feed, particularly for bulls, potentially serving similar purposes in animals. The sedative properties of *Papaver pseudo*-*orientale* and its use even in children highlight a notable aspect of local ethnobotanical knowledge. The direct edible use of its capsules warrants further detailed investigation. Another significant plant identified in this study is *Arum rupicola*, used as a pain reliever during childbirth or unwanted pregnancies. The application of Pistacia species in wound treatment was corroborated with consistent anecdotal evidence from most participants. It was frequently reported that injured wild animals were observed using the stems of these plants to treat their wounds, providing a natural basis for the traditional use of Pistacia species in local medicinal practices.

As outlined in the Introduction and Materials and Methods sections, the local people do not define themselves as a language-based ethnic group such as Turkish, Kurdish, or Zaza. In this multicultural geography, plant names are likely influenced by the Kurdish, Zazaki, Arabic, Turkish, and even Persian languages. It has been observed that plant names in these languages are either used directly or adapted into local names. For instance, in regions where Kurdish and Zazaki are spoken, similar names such as “sir, sire, sirmo, sirmok” are commonly used for wild Allium species. Likewise, participants speaking Zazaki, Kurdish, and Turkish use similar local names for plant genera such as Sambucus, Pistacia, Rhus, Chaerophyllum, Heracleum, Prangos, Cota, Gundelia, Berberis, Cephalaria, Convolvulus, Juniperus, Astracantha, Medicago, Quercus, Geranium, Crocus, Juglans, Mentha, Origanum, Satureja, Thymus, Alcea, Papaver, Hordeum, Rheum, Rumex, Portulaca, Prunus, Pyrus, Rosa, Rubus, Sorbus, Galium, Populus, Tamarix, Urtica, and Eremurus. However, in face-to-face interviews, it was found that certain species, such as *Corylus avellana*, *Celtis tournefortii*, *Equisetum arvense*, *Glycyrrhiza glabra*, *Ficus carica* subsp. *rupestris*, *Paliurus spina*-*christi*, *Prunus mahaleb*, and *Daphne oleoides* subsp. *oleoides*, are referred to using only Turkish or local names derived from Turkish. Conversely, species such as *Scandix pecten*-*veneris*, *Helichrysum plicatum*, *Helichrysum rubicundum*, *Tussilago farfara*, *Xeranthemum annuum*, *Alkanna orientalis*, *Anchusa azurea*, *Silene vulgaris* subsp. *commutata*, *Euphorbia denticulata*, *Euphorbia macroclada*, *Salvia sclarea*, *Malva neglecta*, *Plantago lanceolata*, and *Plantago major* have unique local names not found in other comparative studies. Although these local names are specific to the region, they may also belong to the Zaza language.

## Figures and Tables

**Figure 1 plants-13-02104-f001:**
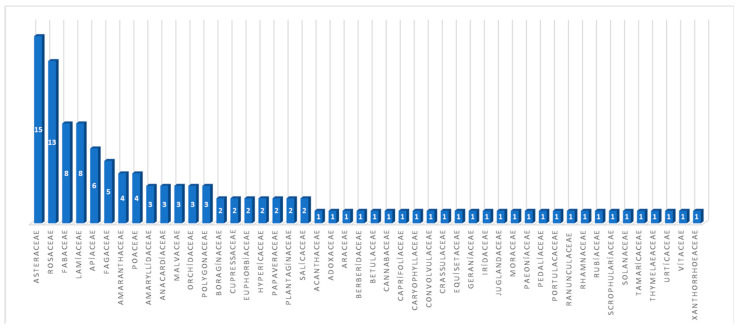
Plant families used as folk medicine.

**Figure 2 plants-13-02104-f002:**
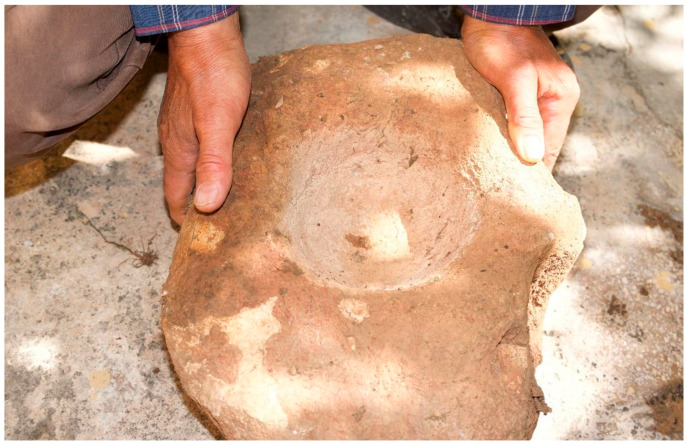
Hand-made stone mortar.

**Figure 3 plants-13-02104-f003:**
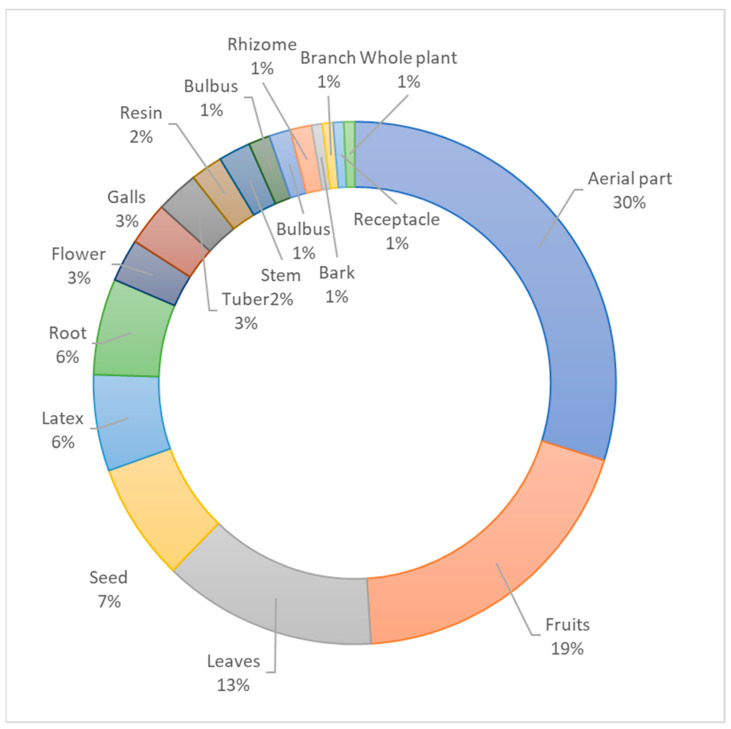
Plant parts used for medicinal purpose.

**Figure 4 plants-13-02104-f004:**
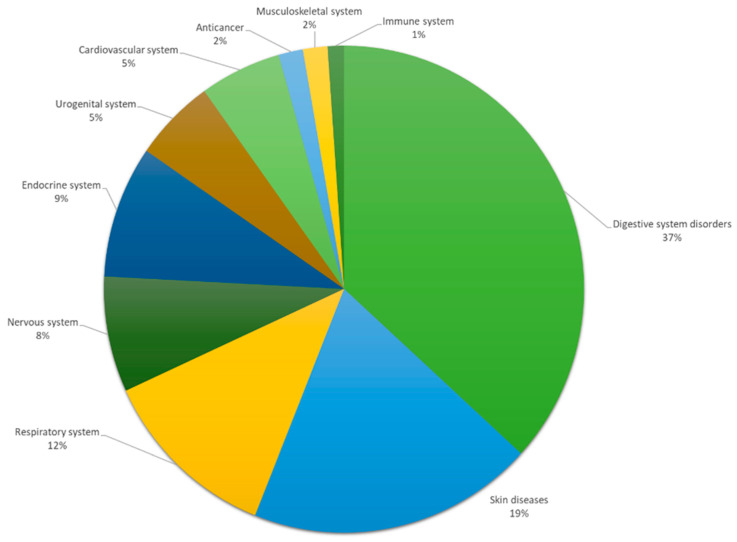
Plants are used to treat the most frequent diseases or disorders in Pülümür.

**Figure 5 plants-13-02104-f005:**
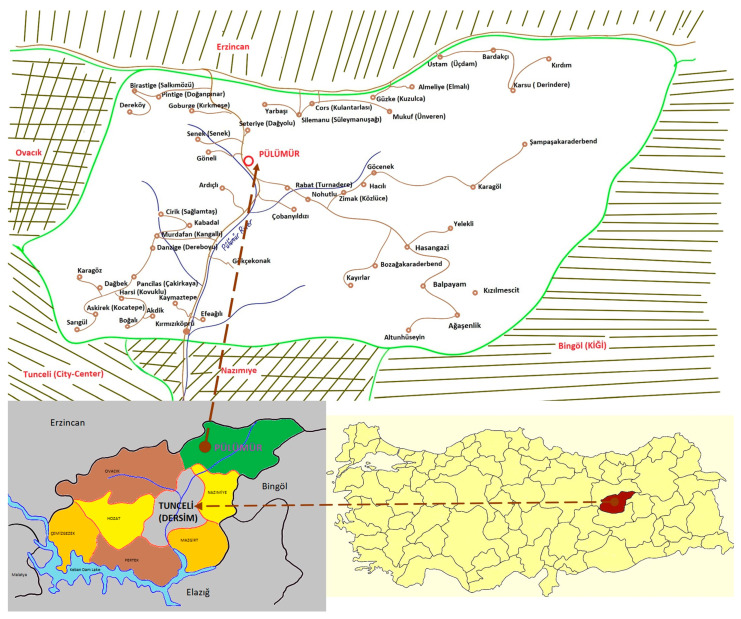
Map of Pülümür and its location in Turkey.

**Figure 6 plants-13-02104-f006:**
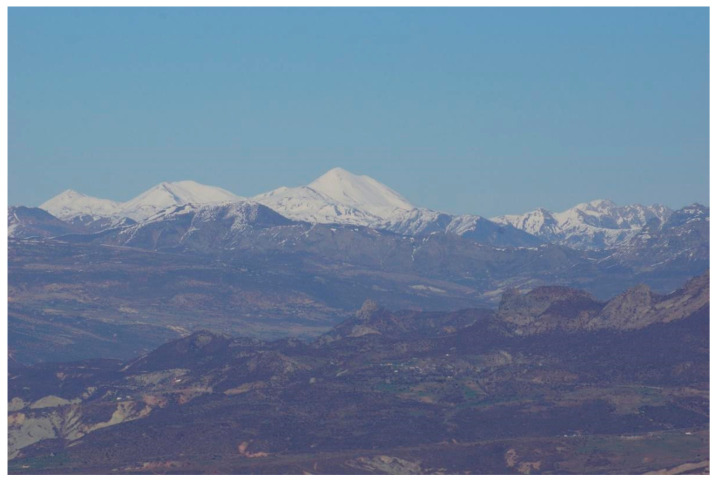
A view of the Munzur Mountains.

**Table 1 plants-13-02104-t001:** Medicinal plants in Pülümür (Dersim-Tunceli).

Family	Scientific Name Voucher Number (MARE)	Vernacular Name	Plant Part Used	Preparation	Utilization Method	Therapeutic Effect/Ailment Treated	Use Value	Use Has Been Reported in Nearby Area and Close Ethnic Cultures
Acanthaceae	*Acanthus dioscoridis* L. MARE17758, MARE20527	Lepe	Aerial part	Infusion	Internal	Diarrhea	0.05	[30,31,32,33]
Adoxaceae	*Sambucus nigra* L. MARE17750	Patpatık, qultıfenk	Fruits	Eaten fresh	Internal	Stomachache	0.05	[34,35,36,37]
				Infusion		Expectoran	0.64	
			Leaves	Infusion		Expectoran	0.05	
Amaranthaceae	*Amaranthus albus* L. MARE20417	Taro unzalık	Aerial part (young)	Cooking	Internal	Constipation	0.45	-
				Decoction		Abdominal pain	0.40	
	*Amaranthus retroflexus* L. MARE20424	Taro sure	Aerial part (young)	Cooking	Internal	Constipation	0.73	[35,38,39,40,41]
	*Chenopodium album* L. subsp. *iranicum* Aellen MARE17717	Sılmastık, Taro luyi	Aerial part (young)	Cooking	Internal	Constipation	0.37	[35,38,41,42,43]
				Decoction		Anthelmintic	0.06	
	*Chenopodium foliosum* Asch. MARE17692	Tuyê mirçiko	Aerial part (young)	Cooking	Internal	Constipation	0.04	[43,44,45]
Amaryllidaceae	*Allium cardiostemon* Fisch. and C.A.Mey. MARE17685, MARE20596	Şirê kutıku	Bulbus	Crushed in stone mortar	External	Ringworm	0.35	-
					Internal	Tuberculosis	0.03	
	*Allium rotundum* L. MARE17712, MARE20607	Şir	Aerial part (young)	Eaten fresh	Internal	Hypertension	0.55	-
	*Allium tuncelianum* (Kollmann) Özhatay B.Mathew and Siraneci MARE17708, MARE20421	Şirê koy	Aerial part (young)	Eaten fresh	Internal	Hypertension	0.95	[37]
				Chewing		Toothache	0.64	
			Bulbus	Chewing		Toothache	0.64	
				Eaten fresh		Hypertension	0.96	
						Insomnia	0.37	
				Crushed in stone mortar	External	Ringworm	0.78	
						Knee pain	0.26	
Anacardiaceae	*Pistacia atlantica* Desf. MARE17709	Qızwan	Resin	-	Internal	Ulcer	0.78	[37,38,39,42,46,47,48,49]
				Mixed with butter	External	Wound	0.91	
			Fruits	Eaten fresh	Internal	Ulcer	0.35	
						Mouth sores	0.27	
	*Pistacia eurycarpa* Yalt. MARE17757	Qızwan, nevzek, vileşk	Fruits	Eaten fresh	Internal	Ulcer	0.71	[30,38,50,51]
						Mouth sores	0.27	
			Resin	-	Internal	Ulcer	0.78	
				Mixed with butter	External	Wound	0.73	
	*Rhus coriaria* L. MARE17722	Zımıq	Fruits	Crushed in stone mortar	Internal	Mouth sores	0.47	[30,31,32,34,35,38,41,43,50,51,52,53,54,55,56]
			Leaves	Infusion	Internal	Diarrhea	0.12	
Apiaceae	*Chaerophyllum crinitum* Boiss. MARE17744	Mendıke	Aerial part (young)	Eaten fresh	Internal	Appetite	0.88	-
	*Eryngium campestre* var. *virens* (Link) Weins MARE20457	Kengerê heru	Stem	Peeling	Internal	Stomachache	0.26	[38,53,57,58]
	*Heracleum cyclocarpum* C. Koch MARE17745	Soy	Aerial part (young)	Eaten fresh	Internal	Appetite	0.36	-
	*Heracleum platytaenium* Boiss. MARE17761	Soy, mendıke	Stem	Peeling	Internal	Appetite	0.78	-
			Aerial part (young)	Eaten fresh	Internal	Hemorrhoid	0.06	
	*Prangos ferulacea* (L.) Lindl. MARE17733	Kınkor	Aerial part (young)	Infusion	Internal	Diarrhea	0.09	[38,39,43,45,47,54,57,59]
			Fruits	Decoction		Cough	0.07	
	*Scandix pecten*-*veneris* L. MARE20441	Poxık	Aerial part (young)	Eaten fresh	Internal	Goiter	0.09	[43]
Araceae	Arum rupicola Boiss. MARE17737, MARE20580	Kardun	Root	Eaten fresh	Internal	Abortive	0.62	[33,51,58]
						Analgesic (during childbirth)	0.81	
			Leaves	Decoction		Constipation	0.03	
Asteraceae	*Achillea vermicularis* Trin. MARE17751, MARE20552	Vasê çegare	Flower	Infusion	Internal	Sedative	0.04	[33,38,44,60]
						Dysmenorrhea (menstrual pains)	0.05	
			Aerial part			Stomachache	0.01	
	*Cota austriaca* (Jacq.) Sch.Bip. MARE17725, MARE17688	Kekowas	Flower	Decoction	Mixed with henna	Headache	0.36	[48,51,61,62]
	*Echinops viscosus* DC. subsp. *bithynicus* (BOISS.) RECH. FIL. MARE17691	Gopıkê heri	Receptacle	Eaten fresh	Internal	Aphrodisiac (male)	0.02	-
						Trouble peeing (Prostatitis)	0.04	
	*Gundelia tournefortii* L MARE20418	Kenger	Aerial part (young)	Eaten fresh	Internal	Appetite	0.78	[30,32,33,34,35,38,40,43,50,53,54,55,56,57,62,63]
			Latex	Chewing	Internal	Gingivitis	0.88	
		Şirik (mature name)	Seed	Crushed in stone mortar	Internal	Appetite	0.19	
	*Helianthus tuberosus* L. MARE17704	Sayê bınê hardi	Root	Eaten fresh	Internal	Appetite	0.93	[35,37,38,41,44,48,60]
				Eaten fresh	Internal	Constipation	0.28	
	*Helichrysum plicatum* DC. MARE20423	Piltan	Aerial part	Infusion	Internal	Kidney stones	0.78	[30,32,35,38,41,48,55,57,60,62,64]
						Diuretic	0.66	
				Decoction		Diabetes	0.09	
	*Helichrysum rubicundum* (K.Koch) Bornm. MARE17714	Piltan	Aerial part	Infusion	Internal	Kidney stones	0.78	-
						Diuretic	0.66	
				Decoction		Diabetes	0.09	
	*Scorzonera latifolia* (Fisch. and C.A.Mey.) DC. MARE17739	Maresungê phepugi	Aerial part (young)	Eaten fresh	Internal	Stomach ulcer	0.27	[33,38,40,44,48,57,60]
			Latex	Mixed with butter	External	Wound	0.94	
	*Scorzonera tomentosa* L. MARE17694	Nerebend, Nune mençuke, Vasê phepugi	Latex	Mixed with butter	External	Wound	0.96	[37,38,57,64]
					External	Burn	0.94	
			Leaves	Eaten fresh	Internal	Stomach ulcer	0.17	
	*Scorzonera veratrifolia* Fenzl MARE17746	Albend, nerebend	Latex	Mixed with butter	External	Wound	0.98	[37,38,57,64]
						Burn	0.81	
						Eczema	0.11	
			Leaves	Eaten fresh	Internal	Stomach ulcer	0.17	
	*Tanacetum cilicicum* (Boiss.) Grierson MARE17711, MARE20432	Kermevas	Flower	Infusion	Nasal Irrigation	Sinusitis	0.15	-
	*Tragopogon buphthalmoides* (DC.) Boiss. MARE20414	Marsunge	Aerial part (young)	-	Internal	Appetizer	0.95	-
			Latex	-	External	Wound	0.39	
	*Tragopogon reticulatus* Boiss. and A.Huet MARE20562, MARE20426	Marsunge	Aerial part (young)	-	Internal	Appetizer	0.95	[35,38,41,57]
			Latex	-	External	Wound	0.39	
	*Tussilago farfara* L. MARE20415	Kersım	Aerial part	Infusion	Internal	Cough	0.28	[32,34,38,43,55,62]
	*Xeranthemum annuum* L. MARE17768	Vasê geji	Aerial part	Infusion	External	Eczema	0.17	[37,38,65]
Berberidaceae	*Berberis crataegina* DC. MARE17707, MARE20572	Qeremux	Fruits	-	Internal	Cold	0.12	[38,57,64]
			Root	Decoction	Internal	Hemorrhoid	0.18	
			Leaves	_	Internal	Diabetes	0.77	
Betulaceae	*Corylus avellana* L. var. avellana MARE17693	Fındıqi	Fruits	-	Internal	Cardiovascular Diseases	0.09	[47]
Boraginaceae	*Alkanna orientalis* (L.) Boiss. MARE17730	Hewazge	Root	Crushed in stone mortar and mix with butter	External	Wound	0.13	[38,40,57]
						Burn	0.15	
	*Anchusa azurea* Mill. MARE17716	Celazon	Aerial part	Infusion	Internal	Diuretic	0.18	[30,32,34,35,38,39,41,43,51,53,56,58,60,62,63]
Cannabaceae	*Celtis tournefortii* Lam. MARE17734	Derdağan, theythavi	Fruits	-	Internal	Diarrhea	0.17	[30,37,38,41,43,51,56,58]
Caprifoliaceae	*Cephalaria procera* Fisch. and Avé-Lall. MARE17696	Gulınge	Resin	-	External	Wound	0.74	[33,38,45]
Caryophyllaceae	*Silene vulgaris* subsp. *commutata* (Guss.) Hayek MARE17697	Tarê voreku	Aerial part (young)	Decoction	Internal	Constipation	0.43	[32,34,35]
Convolvulaceae	*Convolvulus arvensis* L. MARE20483	Perçeke	Root	Maceration	External	Wound	0.04	[32,38,39,43,46,53,63,65,66]
Crassulaceae	*Rosularia sempervivoides* (Fischer ex M. Bieberstein) Boriss. MARE17728	Nune mencuke	Leaves	-	Internal	Appetite	0.69	[45]
Cupressaceae	*Juniperus excelsa* M.Bieb. MARE17701	Terx	Cone	Decoction	Internal	Dysmenorrhea (menstrual pains)	0.51	[38,39,49,62,66]
	*Juniperus oxycedrus* L. MARE17724	Çekem, Merx	Cone	-	Internal	Dysmenorrhea (menstrual pains)	0.43	[32,33,38,40,42,51,53,55]
						Abdominal pain	0.51	
						Shortness of breath	0.65	
				Boiling in water	Inhilation	Upper respiratory infections	0.72	
Equisetaceae	*Equisetum arvense* L. MARE17743	Atkuyruğu	Aerial part	Infusion	Internal	Trouble peeing (Prostatitis)	0.03	[34,38,40,43,48,59,67]
						Rheumatism	0.01	
Euphorbiaceae	*Euphorbia denticulata* Lam. MARE20554	Diliye, Şira ma, Şirokoyi, Vaso şitin	Latex	-	External	Eczema	0.19	[43,44,48,56,60,62]
						Wart	0.29	
	*Euphorbia macroclada* Boiss. MARE20618	Diliye, Şira ma, Şirokoyi, Vaso şitin	Latex	-	External	Eczema	0.19	[32,34,37,38,51,54,55,56,62,63,64]
						Wart	0.29	
Fabaceae	*Astracantha amblolepis* (Fisch.) Podlech MARE17681	Gone	Whole plant	Decoction	Internal	Immune booster	0.08	-
			Root	Chopping	Internal	Aphrodisiac (Male)	0.04	
	*Cicer bijugum* Rech.f. MARE20416	Nukê koy	Seed	-	Internal	Appetizer	0.79	-
	*Colutea cilicica* Boiss. and Balansa MARE20567	Darê avres	Seed	-	Internal	Constipation	0.03	[32,37,38,65]
	*Glycyrrhiza glabra* L. MARE20511	Meyan	Root	Maceration	Internal	Abdominal pain	0.03	[30,38,39,40,41,43,46,47,48,49,51,54,56,58,60,62,63,65,66,67,68,69]
						Antihypertensive	0.10	
	*Lathyrus roseus* Steven MARE17698	Dıldırme	Leaves	-	Internal	Liver disorders	0.13	-
	*Medicago minima* (L.) L. MARE17735	Gurnig	Fruits	Decoction	Internal	Cardiovascular Diseases	0.11	[37,38,63]
	*Trifolium hybridum* L. MARE20551	Nefele	Aerial part	Decoction	Internal	Rheumatism	0.04	-
	*Vicia villosa* Roth MARE20502	Mırzor	Seed	-	Internal	Constipation	0.03	[43,59]
Fagaceae	*Quercus infectoria* G.Olivier MARE20473	Velg	Fruits	-	Internal	Diarrhea	0.17	[31,43,50,56,62,69]
			Galls (Vernacular name is Qanqole)	Crushed in stonemortar	External	Wound	0.05	
	*Quercus libani* G.Olivier MARE20468	Azgılere	Fruits	-	Internal	Diarrhea	0.26	[35,41,51,65]
	*Quercus macranthera* Fisch. and C.A.Mey. subsp. *syspirensis* (K.Koch) Mentsky MARE20529	Velg	Fruits	-	Internal	Diarrhea	0.26	[57]
			Galls (Vernacular name is Qanqole)	Crushed in stonemortar	External	Wound	0.05	
	*Quercus petraea* subsp. *pinnatiloba* (K.Koch) Menitsky MARE17687	Velg	Fruits	-	Internal	Diarrhea	0.62	[33,35,41]
			Bark	Decoction	Internal	Sore throat	0.05	
			Galls (Vernacular name is Qanqole)	Crushed in stonemortar	External	Wound	0.69	
	*Quercus pubescens* Willd. MARE17741	Beru	Fruits	-	Internal	Diarrhea	0.37	[32,65]
			Galls (Vernacular name is Qanqole)	Crushed in stonemortar	External	Wound	0.01	
Geraniaceae	*Geranium tuberosum* L. MARE20590	Xılok	Tuber	-	Internal	Abdominal pain	0.13	[43,53,63,66]
Hypericaceae	*Hypericum perforatum* L. MARE17753	Batov	Aerial part	Infusion	Internal	Ulcer	0.06	[32,34,35,38,39,40,41,42,43,49,55,59,62,67,69]
				macerate in olive oil	External	Wound	0.15	
	*Hypericum scabrum* L. MARE17700	Batov	Aerial part	Infusion	Internal	Ulcer	0.06	[31,32,33,34,35,38,41,43,44,54,56,60,62,64]
				macerate in olive oil	External	Wound	0.15	
Iridaceae	*Crocus biflorus* subsp. *tauri* (Maw) B.Mathew MARE20455	Pivok	Corms	-	Internal	Analgesic (during childbirth)	0.10	[59]
Juglandaceae	*Juglans regia* L. MARE17682	Goze	Seed	-	Internal	Cardiovascular diseases	0.37	[30,31,33,34,35,36,37,38,41,43,46,47,48,50,55,56,57,58,60,62,64,65,66]
			Leaves	Mix with henna Decoction	External	Headache	0.88	
				Decoction	External	Dandruff	0.88	
				Wrapped in a cloth	External	Eczema	0.79	
				Decoction	External	Antifungal	0.87	
				Infusion	Internal	Anthelmintic	0.28	
Lamiaceae	*Mentha longifolia* (L.) L. subsp. *typhoides* (Briq.) Harley MARE17695	Pune	Aerial part	Infusion	Internal	Upper respiratory infections	0.58	[30,32,33,34,35,37,38,39,40,41,42,43,44,45,46,47,48,50,51,54,55,58,60,64]
						Abdominal pain	0.60	
	*Origanum acutidens* (Hand.-Mazz.) Ietsw. MARE17769	Anux, zembul	Aerial part	Infusion	Internal	Upper respiratory infections	0.32	[37,38,41]
						Cold	0.09	
						Carminative	0.67	
			Leaves	-		Abdominal pain	0.95	
	*Salvia sclarea* L. MARE20595	Dime lue	Aerial part	Dried	Internal	Cold	0.32	[38,40,43,49,54,66]
	*Satureja hortensis* L. MARE17749	Kara anux	Leaves	Dried	Internal	Appetizer	0.78	[38,40,43,44,56,62]
						Carminative	0.78	
	*Stachys lavandulifolia* Vahl MARE17715	Vase ça	Aerial part	Infusion	Internal	Cold	0.89	[33,38,39,41,43,44,54]
						Shortness of breath	0.75	
						İmmune booster	0.91	
						Cancer	0.01	
						Insomnia	0.09	
	*Teucrium polium* L. MARE20586	Nanê phepuge	Aerial part	Infusion	Internal	Carminative	0.14	[32,34,35,37,38,39,41,42,43,48,49,51,53,54,55,56,58,60,62,63,64,66]
						Appetizer	0.03	
				Chewing	_	Toothache	0.38	
				Decoction	External	Antifungal	0.24	
	*Thymus kotschyanus* Boiss. and Hohen. MARE17738	Zembulê kemeru	Aerial part	Infusion	Internal	Carminative	0.73	[30,33,35,38,41,43,48,51,60,64]
						Appetizer	0.86	
	*Ziziphora clinopodioides* Lam. MARE17713	Zembulê koyi	Aerial part	Infusion	Internal	Cold	0.64	[37,38,39,43,64,66]
Malvaceae	*Alcea calvertii* (Boiss.) Boiss. MARE17726	Hiro	Aerial part (young)	Infusion	Internal	Cough	0.33	[38,49,67]
	*Alcea dissecta* (Baker f.) Zohary MARE17763	Hiro	Aerial part	Maceration	External	Knee pain	0.27	[37]
			Leaves	Infusion	Internal	Shortness of breath	0.21	
					Internal	Cough	0.18	
	*Malva neglecta* Wallr MARE20442	Vasê veroji	Aerial part (young)	Infusion	Internal	Cough	0.56	[30,32,33,34,35,36,39,40,41,42,43,44,45,48,51,53,54,55,56,57,58,60,62,64,66]
Moraceae	*Ficus carica* subsp. *rupestris* (Hausskn.) Browicz MARE17699	Yabani incir	Fruits	Infusion	Internal	Constipation	0.09	[37,39,43,47,50,51,54,55,56,58,64,68]
			Latex	-	External	Wart	0.88	
						Eczema	0.48	
Orchidaceae	*Anacamptis coriophora* (L.) R.M.Bateman, Pridgeon, and M.W.Chase MARE20501	Vase masen, Sahlep (salep)	Tuber	Infusion	Internal	Cough	0.42	[66]
	*Dactylorhiza osmanica* (Klinge) P.F.Hunt and Summerh. MARE20505	Vase masen, Sahlep (salep)	Tuber	Infusion	Internal	Cold	0.15	-
						Cough	0.55	
	*Dactylorhiza umbrosa* (Kar. and Kir.) Nevski MARE17706	Vase masen, Sahlep (salep)	Tuber	Infusion	Internal	Cold	0.15	[30]
						Cough	0.55	
Paeoniaceae	*Paeonia arietina* G.Anderson MARE17755	Savıle	Aerial part (young)	Maceration	Internal	Diabetes	0.25	[37]
						Cancer	0.05	
				Infusion		Anthelmintic	0.20	
Papaveraceae	*Papaver pseudo*-*orientale* Medw. MARE17740	Xasxasık	Fruits	-	Internal	Sedative	0.44	-
	*Papaver rhoeas* L. MARE17686	Vilıka veyvıke	Aerial part (young)	Infusion	Internal	Cough	0.21	[30,32,34,41,43,53,55,56,58]
					Ear drop	Ear pain	0.07	
Pedaliaceae	*Sesamum indicum* L. MARE17684	Kunci	Seed	Crushed in stone mortar	External	Burn	0.53	[39,43,54]
						Wound	0.53	
Plantaginaceae	*Plantago lanceolata* L. MARE17762	Amenvas	Leaves	Decoction	Internal	Abdominal pain	0.67	[39,43,44,47,48,50,54,56,57,60,62,66,69]
				-	External	Wart	0.73	
						Wound	0.71	
						Eczema	0.07	
	*Plantago major* L. MARE20602	Amenvas	Leaves	Decoction	Internal	Abdominal pain	0.67	[33,34,35,36,37,39,41,42,43,44,45,47,48,51,52,56,57,58,60,62,64]
				-	External	Wart	0.73	
						Wound	0.71	
						Abscess	0.74	
						Eczema	0.07	
Poaceae	*Elymus repens* (L.) Gould MARE20522	Kere şiramok	Rhizome	-	Internal	Diabetes	0.24	[43,67]
	*Hordeum bulbosum* L. MARE20546	Şiramok	Rhizome	-	Internal	Diabetes	0.08	[33,35,41,43,58]
	*Hordeum vulgare* L. MARE20419	Cew	Seed	Crushed in stone mortar and mix with breast milk	Internal	Colic	0.41	[32,39,43,47,51,56,57,64,65]
			Seed	Crushed in stone mortar and mix with egg	External	Wound	0.89	
	*Setaria viridis* (L.) P.Beauv. MARE17721	Şiramo (şiremo)	Aerial part	Infusion	Internal	Kidney stones	0.11	-
Polygonaceae	*Rheum ribes* L. MARE17719	Ribes	Stem	Peeling	Internal	Diabetes	0.83	[30,32,33,34,35,37,39,41,42,43,44,45,46,48,50,52,54,56,57,60,62,66,67,69]
						Carminative	0.46	
						Hemorrhoid	0.27	
	*Rumex acetosella* L. MARE20583	Thırsıka mesin	Leaves	-	Internal	Diabetes	0.80	[32,33,34,35,37,41,43,52]
	*Rumex patientia* L. MARE17679	Thırsıka gau	Leaves	Decoction	Internal	Diarrhea	0.40	[37,42,43]
Portulacaceae	*Portulaca oleracea* L. MARE17748	Pırpar, Taro gilezıng	Aerial part (young)	Decoction	Internal	Constipation	0.54	[30,32,33,35,37,39,41,43,44,47,49,51,52,53,54,55,56,58,62]
						Anthelmintic	0.05	
Ranunculaceae	*Ranunculus fenzlii* BOISS. MARE20446	Adıroke	Aerial part	Decoction	External	Antifungal	0.02	-
Rhamnaceae	*Paliurus spina*-*christi* P.Mill. MARE20507	Karaçalı	Fruits	Decoction	Internal	Kidney stones	0.01	[30,32,34,41,43,48,51,53,54,58,59,62]
Rosaceae	*Cotoneaster ellipticus* (Lindl.) Loudon MARE17703	Nanê milçıkun	Fruits	Decoction	Internal	Anthelmintic	0.03	-
	*Crataegus monogyna* Jacq. subsp. monogyna MARE20523	Guler	Fruits	Decoction	Internal	Cardiovascular Diseases	0.80	[31,32,34,50,55,56,62,67]
	*Crataegus orientalis* Pall. ex M.Bieb. subsp. *orientalis* MARE17752	Şêze	Fruits	Decoction	Internal	Constipation	0.42	[32,33,34,35,41,57,62]
	*Prunus cerasifera* Ehrh. MARE20610	Heruge	Fruits	Decoction	Internal	Diarrhea	0.59	[31,33,43]
	*Prunus dulcis* (Mill.) D.A.Webb MARE17732	Vam	Seed	Decoction	External	Dandruff	0.05	[30,32,39,43,51,55,69]
	*Prunus mahaleb* L. MARE17736	Mehelep (mahalep)	Fruits	-	Internal	Diabetes	0.08	[39,43,53]
	*Prunus microcarpa* C.A.Mey. MARE17764	Mamux	Fruits	-	Internal	Anthelmintic	0.15	[30,43,48,49,50,54,58]
	*Prunus trichamygdalus* Hand.-Mazz. MARE20512	Vame pinç	Seed	-	Internal	Diabetes	0.85	-
	*Pyrus elaeagnifolia* Pall. MARE20573	Muri, şekok	Fruits	Decoction	Internal	Constipation	0.78	[33,55,56]
						Carminative	0.67	
	*Rosa canina* L. MARE17766, MARE20521	Şilan	Fruits	Decoction	Internal	Cold	0.92	[30,32,33,34,35,37,39,40,41,42,43,45,47,48,49,50,51,54,55,56,57,60,62,64,65,67,69]
						Abdominal pain	0.47	
						Anthelmintic	0.19	
	*Rubus canescens* DC. var. *canescens* MARE17747	Dırike	Fruits	Eaten fresh	Internal	Diabetes	0.42	[41]
			Root	Turning into ashes in sheet iron and mix with water	External	Burn	0.73	
						Wound	0.67	
						Eczema	0.09	
	*Rubus sanctus* Shreb. MARE17765	Dırike	Fruits	Eaten fresh	Internal	Diabetes	0.42	[30,31,32,34,35,41,43,51,53,55,56,58,62]
			Root	Turning into ashes in sheet iron and mix with water	External	Burn	0.73	
						Wound	0.67	
						Eczema	0.09	
	*Sorbus umbellata* (Desf.) Fritsch MARE17767	Gileheş, Saa heşi	Fruits	Eaten fresh	Internal	Carminative	0.08	-
Rubiaceae	*Galium verum* L. MARE17760	Gurnig	Aerial part	Decoction	External	Ear pain	0.04	[40,43,45]
Salicaceae	*Populus nigra* L. MARE17705	Qoax	Leaves	Decoction	Internal	Antipyretic	0.48	[57,65]
						Abdominal pain	0.07	
	*Populus tremula* L. MARE17756	Qoaxo pinc	Leaves	Decoction	Internal	Antipyretic	0.38	[34,65]
						Abdominal pain	0.03	
Scrophulariaceae	*Verbascum speciosum* Schrad. MARE17742, MARE20556	Dımê gay	Flower	Infusion	Internal	Abdominal pain	0.02	[43,48]
Solanaceae	*Hyoscyamus niger* L. MARE17689	Bengi	Seed	Burned on a hot iron	Inhilation	Expel mouth, nose, and eye parasites	0.45	[40]
Tamaricaceae	*Tamarix tetrandra* Pall. ex M.Bieb. MARE20609	Saqol	Branch	Burned on a hot iron	External	Eczema	0.03	-
Thymelaeaceae	*Daphne oleoides* Schreb. subsp. *oleoides* MARE17683, MARE20506	Çöpleme	Leaves	-	Internal	Abdominal pain	0.06	[39,66]
				Infusion	Internal	Cancer	0.26	
Urticaceae	*Urtica dioica* L. MARE17729	Dırke, Dırkê mori	Aerial part (young)	-	External	Rheumatism	0.78	[30,32,33,34,35,37,39,40,41,42,43,45,47,48,49,51,53,55,56,57,58,59,60,62,64,66,67,69]
			Leaves	Decoction	Internal	Kidney stones	0.64	
Vitaceae	*Vitis vinifera* L. MARE17723	Silor	Fruits	Decoction	Internal	Carminative	0.27	[35,39,41,43,47,51,53,56,58,62,68]
						Diabetes	0.24	
						Immune booster	0.57	
Xanthorrhoeaceae	*Eremurus spectabilis* M.Bieb. MARE17727	Heluge, gulıke	Aerial part (young)	Decoction	Internal	Anthelmintic	0.24	[33,35,37,41,43,57,64,67,69]
				Crushed in stone mortar	External	Eczema	0.14	

**Table 2 plants-13-02104-t002:** Demographic information.

Age	Gender		Education Level				
	Female	Male	None	Primary	Elemantary	Secondary	Higher
18–30	18	22	0	2	3	11	24
31–45	22	27	0	5	7	19	18
46–60	36	44	3	12	35	16	14
61≥	19	23	19	10	5	5	3
	211						

## Data Availability

All the data relevant to the paper can be found in Table 1.

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
