# Peer review of "Cultural Use and the Knowledge of Ethnomedicinal Plants in the Pülümür (Dersim-Tunceli) Region"

_plants, 2024, doi:10.3390/plants13152104_

Round 1

Reviewer 1 Report

Comments and Suggestions for Authors

A very interesting work, with a lot of new findings, which were very carefully collected and analyzed. Unfortunately the English in the introduction is a bit choppy. E.g.: "Turkey has a remarkable biodiversity due to its geographical location, climate, water 29

resources and geomorphological diversity. The richness of biodiversity in plants is deter- 30

mined by the number of endemic plants as well as the number of plant taxa growing in 31

that region. Anatolian diagonal is very rich terms of endemic plants and it is the most 32

important region in plant biodiversity in Turkey. [1-4] This study was conducted in the 33

Munzur Mountains which is a part of this diagonal and located in the Pülümür region." 

What is Anatolian diagonal? Does the statement in the sentence refer to Turkey as a whole or to the region being examined? Not understandable.

Comments on the Quality of English Language

Translation from Turkish to English is a bit bumpy

Author Response

A detailed explanation has been made about the Anatolian diagonal. Requested English Proofreading were made

Reviewer 2 Report

Comments and Suggestions for Authors

Overall Comments

 The manuscript provides valuable ethnobotanical knowledge regarding medicinal plants in the Pülümür region (Turkey). The data collected have the potential to offer relevant insights into the local ethnobotany of the area. However, the study requires significant improvements before it is ready for publication. The authors need to clearly define their research questions, justify them beforehand, and then explicitly link these questions with their data analysis and results. The objectives of the study should be better introduced by expanding the introduction to explain their importance. All research inquiries should be justified and explained. For instance, why did the authors explore the differences and similarities between their results and uses found in other areas? Why did the authors collect informants education levels and ages?

 The sampling strategy should be explained and defined according to the research questions, and the analysis should address these questions. Currently, it seems the authors aimed to explore several aspects related to the intra- and intercultural variation of ethnobotanical knowledge regarding medicinal plants but did not clearly explain this. The manuscript suggests that the authors consider age, gender, and education of the informants, but it does not clearly analyze the potential differences. Additionally, information on the differences and similarities in plant uses between the authors data and data from other areas should be detailed in the results section. Thus, the methods need improvement.

 Finally, the manuscript should be reviewed to eliminate repetitions and enhance style (many typographic errors are present). There are several places where information is repeated, such as between the introduction and the study area section, and between the methods and the results. The manuscript will be clearer if repetitions are deleted, and the space is used to develop the methods and results. English proofreading is necessary.

 Introduction

 The introduction provides an interesting background on the specific region where the study takes place. However, before delving into this specific area, the introduction needs an initial paragraph or two that justifies the aim of the study within a broader context. This could be framed by questions that justify the authors research objective, such as: Why is it important to study medicinal plants and their uses in general? Why is it important to document this knowledge corpus? What is the rationale or relevance of exploring similarities and differences between different regions?

Results

 The results should be better organized according to the research questions explored. Figure 1 is difficult to read and should be improved for clarity. While Table 1 seems to contain all the information collected in this study, this information should be better presented within the text. Additionally, Table 1 should specify the language of the vernacular names, as the authors collected local names in different languages.

 The statement "It was observed that females were more competent in the use of herbs than males" needs supporting data. What kind of data were used and analyzed to support this statement? Why did the authors collect the education levels of their informants? What is the rationale behind this?

 Figure 3 should be formatted to include all the parts used; as it stands, it lacks several of them.

Materials and Methods

 The manuscript lacks ethical information. Regarding ethical procedures: Did the authors obtain free prior informed consent? How did they handle confidentiality, anonymity, and intellectual property?

Figure 5 is informative. However, the appendices are missing, which makes it difficult to evaluate the rigor of the methods used. Line 237 repeats information already provided in line 217 regarding "89 days between May 2016 and September 2019."

 The sampling strategy needs clarification. How did the authors determine whether a person was knowledgeable? The data collection process is unclear. Did the knowledge regarding plants come directly during the trips with the interviews? Did all the interviewees guide the authors and show them the plants and then explain their uses (such as the walk-in-the-wood method)? Was a different method used? This needs to be better explained.

 "Statistical data analysis in this study was conducted following guidelines for best practices in ethnopharmacological research." Please provide the reference for this.

Conclusion

 The statement, In conclusion, the study clearly demonstrates that plants continue to play an important role in people's basic health care in the Pülümür district, requires more data to support it. More information is needed about what informants are doing and using as treatment. Knowing plant uses does not mean people are actively using them. The research shows that "knowledgeable" people in this area still hold knowledge about medicinal plants.

Line 272, correct the name to Pülümür population.

References

 Homogenize the format of the references according to the authors guidelines. Currently, there are several discrepancies in the references list. Clarify the nature of Ref 17.

What the research brings is that knowledgeable people from this area still hold knowledge on medicinal plants.

 Line 272, correct the name Plümür population

Comments on the Quality of English Language

The manuscript should be reviewed to eliminate repetitions and enhance style (many typographic errors are present). There are several places where information is repeated, such as between the introduction and the study area section, and between the methods and the results. The manuscript will be clearer if repetitions are deleted, and the space is used to develop the methods and results. English proofreading is necessary.

Author Response

Introduction

A paragraph and research questions were added to the introduction.

Results

A short paragraph explaining table 1 was added to the conclusion section. However, the language of the local names of the plants could not be included in the table. The reason why this could not be added is that there are many cultures in and around the region such as Kurdish, Turkish, Zaza and Arab. Etymological research of plant names may also be the subject of another study.

A paragraph was added to the conclusion section explaining differences in usage based on age, education and gender.

figure 3 corrected

Materials and Methods

Lack of ethical information has been resolved. Appendices added. The sampling strategy is explained. "Statistical data analysis in this study was conducted following guidelines for best practices in ethnopharmacological research." added reference

Conclusion

The requested correction was made in the conclusion section.

References

References were arranged in ACS format using Endnote.

Requested English Proofreading were made